# Initial Positive Indications with Wearable Fitness Technology Followed by Relapse: What's Going On?

James Parker * 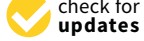, Urban Johnson and Andreas Ivarsson

Center of Research on Welfare, Health and Sport, Halmstad University, 301 18 Halmstad, Sweden;
urban.johnson@hh.se (U.J.); andreas.ivarsson@hh.se (A.I.)
* Correspondence: james.parker@hh.se

**Abstract:** The motivational influence of wearable fitness technology (WFT) on increasing physical activity (PA) is unclear, and improvements in PA have been shown to be driven by both intrinsic and extrinsic motivation. In the current study, PA (daily number of steps), moderate to vigorous intensity physical activity, and muscular strength training were measured over 6 months on, originally, 16 randomly selected sedentary community workers (mean age = 51 years). Moreover, self-determined motivation (Behavioral Regulation in Exercise Questionnaire-2) was measured before, midway, and after a 6-month intervention program that included motivational interviewing, as well as the use of WFT and a structured outdoor gym program. Our findings showed WFT, in combination with motivational interviewing, initially helped the participants meet recommended guidelines for PA in terms of at least 10,000 steps per day, and at least 150 min of moderate aerobic activity per week. There was a large decrease in participants' PA and increase in introjected motivation between the first half (3 months) and the second half of the intervention (6 months). The increase in introjected motivation suggests that toward the end of the 6-month intervention, participants engaged in PA to satisfy external demands or avoid guilt, which may lead to less-persistent behavior change.

**Keywords:** health; motivation; physical activity; wearable fitness technology

## 1. Introduction

Healthy living is associated with positive outcomes, such as high levels of psychological and social well-being, physiological and metabolic health, physical health, and cognitive functioning [1]. A major challenge in most societies is helping citizens increase physical activity (PA) levels to help lower healthcare costs. Inadequate PA is one of the four leading behavioral risk factors for noncommunicable diseases worldwide [2]. Adults are recommended to engage in a minimum of 150 min of moderate-intensity PA, 75 min of vigorous-intensity PA, and two or more days a week of muscle-strengthening activities [3]. Numerous intervention programs, aimed to increase physical activity in sedentary people, have been developed and tested [4]. Outdoor exercise interventions are, in general, related to increases in PA [5], and access to outdoor exercise equipment can help increase activity levels in people who do not usually exercise [6]. Intervention studies have suggested that open-air environments placed in urban green areas, may have direct and positive impacts on mental health and well-being (e.g., Barton and Pretty [7]; Johnson et al. [8]). More research concurrently investigating outdoor PA, moderate to vigorous physical activity (MVPA), and muscular-strengthening activities over an extended period (e.g., >3 months) is needed.

The public health implications of using activity-tracking devices to promote and monitor behavior change and increase physical activity are quite promising [9]. The use of activity-tracking devices has, for example, been found to increase PA and PA motivation [10]. Most activity-tracking devices offer immediate feedback tied to goals (e.g., 10,000 steps) and tracking changes in PA can motivate steady progress toward goals and

increased self-efficacy. Activity-tracking devices have been suggested to have the potential to revolutionize PA research by allowing, for example, real-time data to be gathered [9]. It is still unclear whether it is the tracking device itself or the intervention that leads to positive outcomes. Activity devices are considered financially economical for research because of the reasonable cost, but may not be equally accessible to the general population, such as those with a lower social economic background [11]. One solution to this inequality is activity trackers being prescribed by doctors as an economically viable option and successful component in healthcare interventions.

The application of digital health software and devices, such as wearable fitness technology (WFT; e.g., smartwatches) and smartphone applications (apps), have the potential to help people increase their levels of PA [12]. There are a number of key factors that influence behavioral intention to use health technologies that include perceived ease of use and perceptions of effectiveness [13]. There also are some barriers to the uptake of WFT, such as the level of app literacy among end users [14]. Previous research [15] found that feedback from WFT can lead to users becoming more goal-oriented with increased feelings of personal control. Such changes seem beneficial and can increase feelings of self-efficacy [9]. Results from recent studies, however, are mixed. Schiel, Kaps, and Bieber [16] showed improvements in intrinsic and extrinsic motivation, and another study on adolescents reported that short-term increases in motivation were driven by feelings of competition, guilt, and internal pressure [17], and these feelings were unlikely to result in persistent and sustainable behavior change. Research that investigates the motivational influence of WFT would contribute to an evidence base on the influence of WFT on health.

Self-determination theory (SDT) is an organismic theory of human motivational processes [18] that provides an understanding of the initiation and maintenance of physical activity. Different types of motivation (intrinsic, identified, introjected, external, and amotivation) are proposed to exist along a continuum ranging from lower to higher levels of self-determination. The different types of motivation range from undertaking an activity for the inherent pleasure, to engaging in a behavior to avoid punishment or obtain a reward [19]. According to SDT, individuals are most effective and persistent in pursuing a healthy lifestyle when they are intrinsically motivated [19]. A logical implication for health practitioner would be to find strategies to aid individuals' in finding PA intrinsically satisfying or to truly identify with and value the outcomes of PA. For example, pleasant environments that surround exercise settings (e.g., parks) have been suggested to indirectly increase motivation to exercise [8]. WFT, however, provides external rewards, such as achievement notifications, and may enhance extrinsic motivation for behavioral change and PA.

Professional coaching via weekly information sessions has been shown to encourage healthy behaviors [20]. Building on the SDT framework, one approach that has been effective to support behavior change is motivational interviewing (MI) [21]. In MI, the interaction between the counselor and client is based on collaboration, non-judgment, and autonomy [22]. MI targets the three key components in self-determination theory, and this approach has been found to be effective in terms of promoting behavior change [23]. There is evidence that interventions using MI-inspired techniques, often including weekly or biweekly sessions, support behavior change and PA [24]. Nevertheless, there is limited research on the combined effects of WFT and MI on behavior change and PA.

Given the positive effects of physical activity, MVPA, and muscular-strengthening activities on health, understanding the potential influence of WFT and face-to-face coaching on increasing PA, behavior change, and motivation would seem important. The purpose of this study was to investigate the influence of a 6-month intervention using health and lifestyle technology, in the form of a WFT and smartphone app designed to encourage muscular strength training, in combination with motivational interviewing on behavioral and motivational outcomes.

## 2. Materials and Methods

### 2.1. Participants and Inclusion Criteria

The inclusion criteria used in the selection of the participants were: (a) have a primarily sedentary job; (b) train less than once a week; (c) be employed within Halmstad City Council; (d) live (have residence) in Halmstad; and (e) have access to a smartphone. A random selection of participants from a pool of 66 volunteer participants, in which a weighting of gender was carried out due to an overbalance of women, resulted in 20 (male = 7; female = 13) with a mean age of 48.5 years (SD = 9.9); men 42.4 (SD = 9.9) and women 51.7 (SD = 8.6). Four participants dropped out during the study, resulting in 16 participants (male = 5; female = 11) with a mean age of 51 years (SD = 8.4) and body mass index (BMI) of 28.23 (SD = 3.79); men 45 years (SD = 10.4) and BMI of 31.02 (DS = 2.54); and women 53.7 (SD = 6.1) and BMI of 27.00 (SD = 3.67). Figure 1 shows the flow of participants through the intervention.

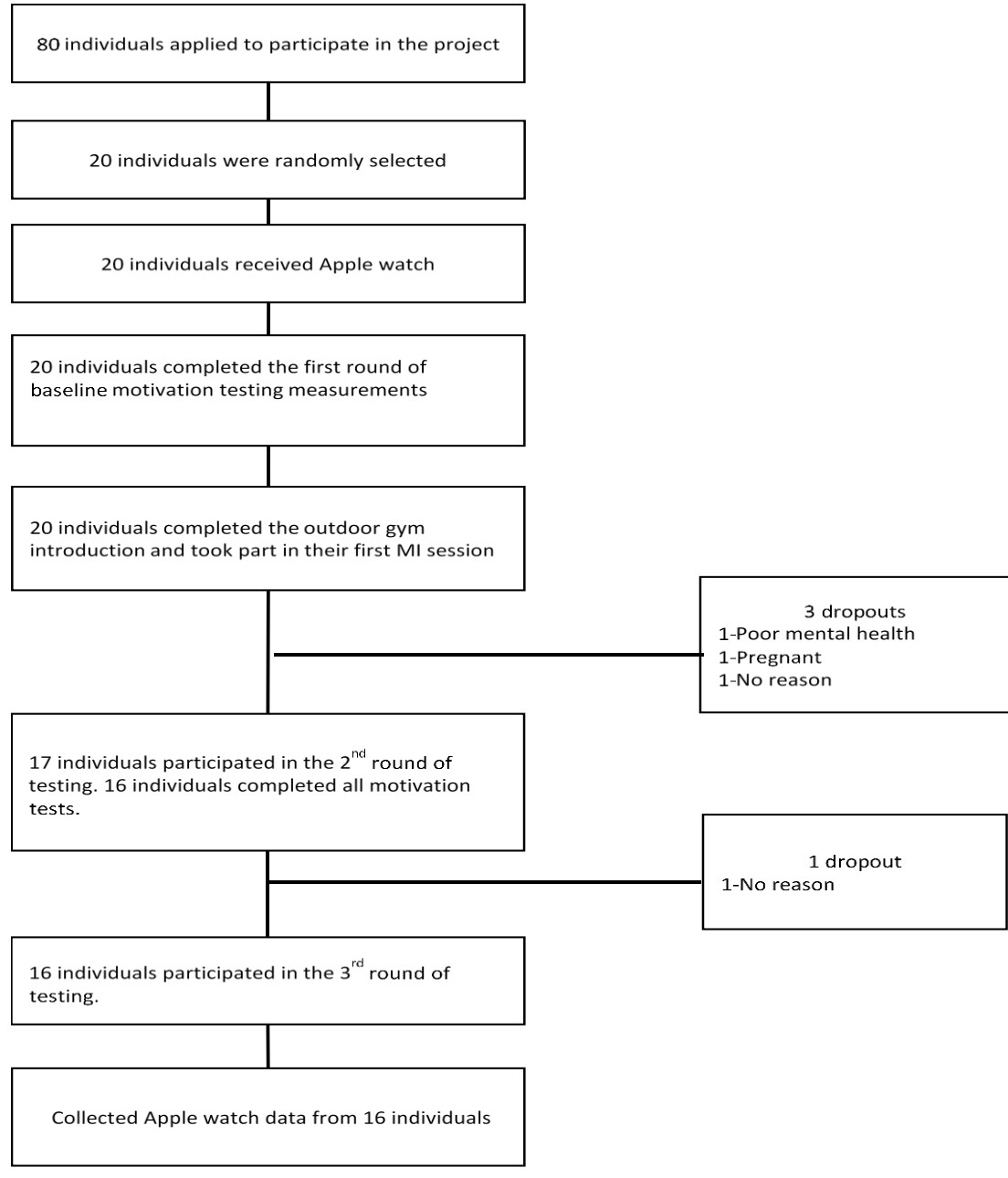

**Figure 1.** Flow of participants through the intervention.

### 2.2. Motivation Measurements

Data were collected using the Behavioral Regulation in Exercise Questionnaire-2 (BREQ-2) [25] to measure the psychological construct of motivation regulations. The BREQ-2 contains 19 items (e.g., "It's important to me to exercise regularly") measured on a 5-point Likert scale ranging between 0 (not true for me) to 4 (very true for me). The scale measures five motivational regulations: amotivation (4 items, McDonald's $\omega$ = 0.89), external (4 items, McDonald's $\omega$ = 0.93), introjected (3 items, McDonald's $\omega$ = 0.72), identified (4 items, McDonald's $\omega$ = 0.81), and intrinsic motivation (4 items, McDonald's $\omega$ = 0.89).

### 2.3. Physical Activity Intervention

The participants took part in the pre-post intervention design aimed to increase PA and well-being (see Table 1). There were three data collection occasions: the first week (the week after the participants received the WFT but before the introduction to the outdoor gym and MI sessions), after 3 months, and at the end of the intervention 6 months after the first weeks test. The first week was defined as the period between receiving the smart watch until the first outdoor gym session. All participants received a Smartwatch (Apple Watch1) and were instructed on how to use the basic functions of the activity sensor (steps, active calories, time, and synchronization with the iPhone).

**Table 1.** Time plan for the study.

| Month | Activities |
|---|---|
| First week | Distribution of smartwatches |
| 0 | Beginning of the intervention<br>Introduction to the outdoor gym<br>1st psychological questionnaire<br>1st motivational interviewing session. |
| 3 | 2nd psychological questionnaire<br>2nd motivational interviewing session. |
| 6 | End of the intervention<br>3rd psychological questionnaire<br>Apple watch and health app data collection |

### 2.4. Motivational Interviewing

PA was supported through two individual motivational interviewing coaching sessions, and each participant received a resistance-training program designed for use in an outdoor gym. The individual motivational interviewing coaching session was composed of four processes [22]. In the engaging process, a connection was established and rapport was built. After that, the process of focusing was developed to maintain a detailed direction in the conversation about change to support exercise behavior. The next process was evoking, which involved the participants' own motivations for change. The last process was planning, which involved both developing commitment to change and formulating an action plan.

All participants were given access to an app (ParkStark), specially constructed for the study and designed to encourage muscular strength training, and they were encouraged to use ParkStark when training at the outdoor gym. The ParkStark app contained a resistance-training program designed for use in an outdoor gym, along with exercise descriptions for all exercise machines at the gym, and information on how often the app was used. Throughout the 6-month intervention period, each participant was advised to include muscular strengthening activities 2–3 days a week. When the intervention started, the participants were introduced to an outdoor gym and instructed on how to use it (instructors were present at the start of the intervention for each participant) to further promote sustainable physical activity. Moreover, they were advised to track their activities

through the default functions on their watches. During their exercise, it was possible for the participants to access direct feedback on their physical activities at the gym (e.g., heart rate, calories burned, session duration).

*2.5. Procedure*

Figure 1 outlines the time plan for the study procedures from the first contact with the participants until the final testing session 6 months later. Ethical approval for the study was granted by the regional ethics committee (reference number 2016/843).

*2.6. Data Reduction*

Data were first stored locally on the smartphone and then downloaded from the Health Data app on smartphones. The data file was then run through a script in Python 3.7.2 (health data reader parser; Python Software Foundation, version 3.8, Wilmington, DE, USA), through which data for steps, MVPA time, heart rate (HR), and ParkStark were extracted and stored as separate files. All the extracted data included timestamp information that was then used to aggregate the data to hourly data points throughout the 6-month intervention period. All daily measurements with less than 8 h of recorded heart rate data were excluded.

Geographical locations of the participants' phones were recorded when they used the ParkStark app to track not only when the ParkStark app was used, but also when it coincided with muscular-strengthening activities in an outdoor gym. Participants were considered to have used the outdoor gym when the start and finish of the session had a geographical location within a 100 m radius of the gym.

*2.7. Data Analysis*

A repeated-measures analysis of variance (ANOVA) assessed any differences (pre–mid–post) for each physical activity and motivation variable. A *p*-value $< 0.05$ was considered statistically significant in all analyses. Eta squared ($\eta^2$) was used to assess the between-measurement effect sizes. Effect sizes of 0.01–0.06 were considered small in magnitude; those in the 0.06–0.14 range were medium, and those above 0.14 were large [26]. Post hoc differences were analyzed using paired *t*-tests and Cohen's effect size d for repeated measures. A d of 0.20–0.50 was considered small in magnitude; those between 0.50–0.80 were medium, and those above 0.80 were large [26].

## 3. Results

The repeated-measures ANOVA (Table 2) showed statistically significant differences and large effect sizes for daily steps ($p = 0.02$, $\eta^2 = 0.25$), daily exercise time ($p = 0.01$, $\eta^2 = 0.30$), outdoor gym visits ($p = 0.01$, $\eta^2 = 0.54$), and introjected regulation ($p = 0.01$, $\eta^2 = 0.26$). The subsequent post hoc analyses confirmed statistically significant differences and a large reduction for MVPA ($p = 0.02$, d = 0.87) between months 0–3 and months 4–6, and a large reduction for steps ($p = 0.02$, d = 0.80) between months 0–3 and months 4–6. The post hoc analysis of introjected regulation showed a statistically significant large increase ($p = 0.006$, d = 0.97) between the first week and the 6-month follow-up (Tables 2 and 3). There were substantially skewed distributions around the averages in a number of the measures, in particular, app usage and the number of visits to the outdoor gym, for which the standard deviation was larger than the average number of uses (Table 4). For the ParkStark app, in the first half of the intervention, the range of use was from 0 to 32 times. In the second half, nine participants did not use it at all, and the use for the other participants ranged from twice to 25 times.

**Table 2.** Differences in repeated-measure *p*-values and effect sizes in physical activity and motivation measures. Number of participants = 16.

|  | *p* | $\eta^2$ |
|---|---|---|
| Physical activity |  |  |
| Steps | 0.017 | 0.251 |
| Exercise | 0.009 | 0.303 |
| Outdoor gym (visits) | 0.002 | 0.536 |
| Strength training (APP) | 0.083 | 0.187 |
| Motivation measures: BREQ-12 |  |  |
| Amotivation | 0.370 | 0.069 |
| External regulation | 0.520 | 0.046 |
| Introjected regulation | 0.014 | 0.264 |
| Identified regulation | 0.486 | 0.050 |
| Intrinsic regulation | 0.893 | 0.008 |

**Table 3.** Descriptive statistics for motivation measures. Number of participants = 16.

|  | Month 0 Mean (SD) | Month 3 Mean (SD) | Month 6 Mean (SD) |
|---|---|---|---|
| Amotivation | 1.2 (0.3) | 1.2 (0.4) | 1.2 (0.5) |
| External regulation | 1.3 (0.7) | 1.4 (0.8) | 1.3 (0.8) |
| Introjected regulation | 2.1 (0.9) | 2.6 (1.2) | 3.0 (0.9) |
| Identified regulation | 3.9 (0.8) | 4.0 (0.5) | 4.0 (0.7) |
| Intrinsic regulation | 3.7 (0.9) | 3.7 (0.9) | 3.6 (1.0) |

**Table 4.** Descriptive statistics for physical activity. Number of participants = 16.

|  | First Week Mean (SD) | Month 0–3 Mean (SD) | Month 4–6 Mean (SD) |
|---|---|---|---|
| Physical activity (steps per day) | 14,927 (7925) | 12,914 (4486) | 11,407 (4486) |
| MVPA * (minutes per day) | 34.9 (18.5) | 35.3 (20.0) | 22.5 (11.4) |
| Total amount of app usage | 0 | 8.4 (10.1) | 6.0 (8.9) |
| Total number of outdoor gym visits | 0 | 4.2 (3.9) | 0.9 (2.0) |

* MVPA = moderate to vigorous physical activity.

## 4. Discussion

The current study sought to investigate the influence of a 6-month intervention using health and lifestyle technology on behavioral and motivational outcomes. The findings indicated that during the first three months of the intervention, participants on average met, or exceeded, the recommended guidelines for PA in terms of 10,000 steps per day, and at least 150 min of moderate to vigorous aerobic activity per week, but not two muscle-strengthening activities a week (average = 0.7 sessions a week). This initial positive indication was followed by a relapse when participants showed large (ES) reductions in measures of PA and an increase in introjected motivation between months 0–3 and months 4–6.

The app designed to encourage strength training at an outdoor gym was used more often than the participants used the outdoor gym (Table 4). These results suggest that the app was, perhaps, more successful than the outdoor gym at motivating participants to engage in strength training. Our results showed only a few participants used the app designed to encourage muscular strength training, but these participants used the app once a week or more. A recent study by Peng et al. [14], showed that a potential reason for nonadoption of health apps was low app literacy among end users, particularly among older individuals, and this finding could, in part, explain the low adoption rate of our strength-training app. Some exercise-intervention studies (e.g., Fortier et al. [24]) have shown, from a self-determination theory perspective, that social interaction can support basic psychological needs such as competence and relatedness and, in turn, enhance

intrinsic motivation. We speculated that participants who used the app, but not in the vicinity of the outdoor gym, found strategies to perform muscle-strengthening activities that better suited their individual circumstances, such as training at home or at another gym with colleagues or family members. Finally, the aim of MI sessions was to promote participants' sense of ownership over their own behaviors. The MI sessions included encouraging choice and providing a menu of options for behavior change, of which strength training was one such choice. Participants may have simply made other choices to increase PA and improve their health and wellbeing.

One finding to explain some of the reduction in all forms of PA during the second half of the intervention was the increase in introjected motivation among the participants. Our results showed a large increase in introjected regulation from the first week to the final measurement, suggesting that participants engaged in the outdoor activity to satisfy external demands or to avoid guilt and shame. According to the SDT, individuals are most effective and persistent in pursuing healthy lifestyles when they are intrinsically motivated [19]. In the current study, self-monitoring and MI were used to encourage behavior change and enhance PA levels. Due to the current study's design, we were unable to show if WFT and MI influenced motivation in different ways. Results from a recent 3-month outdoor physical activity study showed, however, that the group who received WFT and MI coaching maintained introjected regulation levels when compared to the group who only received WFT, who showed a decrease in identified regulation, suggesting MI coaching helped to maintain a somewhat more internal motivation [8]. Although WFT and healthy-lifestyle apps are useful tools for self-monitoring PA levels, it is unclear how they influence motivation. There is a growing body of evidence that WFT may increase PA by increasing extrinsic motivation. Kerner and Goodyear [17] reported that participants who had other people monitor their progress presented a risk of disappointing others and the potential of causing them embarrassment and feeling judged. The constant measurement by WFTs can draw a user's attention to the outcome and undermine intrinsic motivation by making activities feel less enjoyable [27]. The increase in introjected regulation in our results may, in part, explain the large decrease in PA.

There were some limitations to the current study that were generic to investigations aiming to increase physical activity. For example, despite our effort to recruit a gender-balanced pool, about two-thirds of the participants were female; the gender-balance trend has previously been reported [14], and reflects the population at large in which more men are physically active than women, and so therefore there were more women who were eligible for this study. Another limitation to our study design was the time period (3 months); motivation was measured and previous research [17] has reported that the novelty effects of WFT become apparent after four weeks, which may suggest that the reductions occurred much earlier than we showed. Our study showed no significant change in PA or MVPA measurement between the first week and the first half of the intervention (month 1–3), which suggests the novelty may not have become apparent after only four weeks. Our results, however, showed a small effect size ($d = 0.44$) and a nonsignificant ($p = 0.34$) difference between measures for PA (steps) at the first week and months 0–3, but not for MVPA, which may indicate the effect of novelty on PA (steps) but not MVPA.

## 5. Conclusions

The public health implications of the growth in outdoor exercise areas and WFT are encouraging, although the economic cost of WFT may mean that these devices are not accessible to people with a lower socio-economic background. One strength in our intervention design that could be implemented in healthcare settings as a means of ensuring access to those with a lower social-economic background was only two MI sessions, access to a community outdoor gym, and WFT and an App that could be made available by healthcare providers. Previous research highlights that it is still unclear how WFT alone influences long-term changes in PA behavior. The 6-month data collection period and the first week of baseline PA prior to the commencement of the intervention gave some insight

into this, and we were able to show the high initial levels of PA and MVPA influenced by the watch and the onset of the intervention and the following relapse. Adults are recommended to engage in two or more days a week of muscle-strengthening activities [4], and studies using an observational design have reported that outdoor gyms increase exercise behavior [28]. In the current study, we followed how often the participants used the outdoor gym, and showed that they found other strategies to perform muscular-strength training than using the outdoor gym.

The SDT provides an understanding of the motivational regulations underlying the initiation and maintenance of physical activity. Findings in our study identified increase in introjected regulation and maintenance of autonomic regulation among participants. This may explain the reduction in PA levels, which nevertheless stayed above the recommended levels of PA, during the latter part of the study, and supports the use of the SDT in a PA and MVPA context. Furthermore, the self-determination theory perspective, that social interaction can support basic psychological needs such as competence and relatedness, gives an idea of how to continue to apply the SDT model in order to gain a deeper understanding of outdoor exercise, in particular the muscular-strength-training behavior.

The current study provides some practical implications that may help practitioners develop WFT and smartphone apps designed to encourage muscular-strength training. Practitioners should consider how to best combine technologies and coaching without increasing extrinsic motivation that likely leads to less-sustainable behaviors despite the initial short-term positive changes in PA, MVPA, and muscular-strength training. A second implication for local governmental agencies, which provide outdoor strength-training equipment, is that access to an app designed to encourage muscular-strength training is probably an inadequate intervention.

Our intervention program did not, on the whole, motivate the participants to use the outdoor gym, and to better understand how to support long-term improvement in PA, more studies that investigate changes in PA, MVPA, and muscular-strength training in relation to novelty and nonadoption are needed. Future research should consider investigating how MI sessions can encourage muscular-strength training and app utilization, and how environments that support basic psychological needs such as competence and relatedness influence outdoor gym usage.

**Author Contributions:** All authors contributed to study planning, interpretation of results, drafting, and finishing the manuscript. U.J. and A.I. were responsible for the MI sessions. J.P. was responsible for data collection and data analysis. All authors have read and agreed to the published version of the manuscript.

**Funding:** The study was financed by a grant from The Knowledge Foundation, Sweden (grant number 20160097).

**Institutional Review Board Statement:** The study was conducted according to the guidelines of the Declaration of Helsinki, and approved by the Regional Ethics Board, Lund (reference number 2016/843).

**Informed Consent Statement:** Informed consent was obtained from all subjects involved in the study.

**Data Availability Statement:** The data used for this study are available on request from the corresponding author.

**Acknowledgments:** The authors are particularly grateful to Erik Blomberg and Camilla Schough at Eleiko Sport AB, Sweden; Erik Viberg, Anton Bärwald, and Pelle Wiberg at Swedish Adrenaline, Sweden; Sofia Warpman at Halmstad Municipality, Sweden; and Ingrid Svetoft and Mark Andersen for their constructive feedback and valuable input throughout the entire research project and this study.

**Conflicts of Interest:** The authors declare no conflict of interest.

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
