# Peer review of "Initial Positive Indications with Wearable Fitness Technology Followed by Relapse: What’s Going On?"

_sustainability, doi:10.3390/su13147704_

Round 1

Reviewer 1 Report

The article address an interesting and important topic of the present societies: the importance of Wearable Fitness Technology.

The article is well structured, the analysis is well done and the results are clearly presented.

The recommendations refer to a broader understanding of Wearable Fitness Technologies within the general framework of future improvement of human well being as a result of adopting the technological progress. The purpose of the article is indeed to present a specific case but the introduction and conclusions could be improved by adopting a more general view of the topic.

Author Response

Manuscript ID: sustainability-1214606

Title: Initial Positive Indications with Wearable Fitness Technology Followed by Relapse: What’s going on?

We would like to take this opportunity to thank reviewers for valuable and relevant comments that have increased the overall quality of our revised manuscript. Below you will find our suggestion for feedback on comments made. We hope that our answers satisfy your critical comments and that you consider accepting our revised manuscript.

Sincerely,

the authors

Reviewer 1

Reviewer’s comments

Authors’ Responses

The recommendations refer to a broader understanding of Wearable Fitness Technologies within the general framework of future improvement of human well being as a result of adopting the technological progress.

The purpose of the article is indeed to present a specific case but the introduction and conclusions could be improved by adopting a more general view of the topic

Line 32-52: we have added a more general view of the topic to the introduction.

Line 264-279 we have improved the discussion by including a general view of the topic

Reviewer 2 Report

The Article “Initial Positive Indications with Wearable Fitness Technology Followed by Relapse: What’s going on?” requires several changes before it will be published. There are some remarks concerning this article:

  1. The title of the article is not correctly formulated.
  2. There was not presented hypothesis.
  3. Some terms usage could be corrected, e.g., “physical capacity” etc.
  4. I missed some anthropometric data in this research (e.g., body weight, height, BMI etc.) it would be more informative and additionally for trying to find some links to the motivation. It would be possible to add this information, because in apple watches this information necessary at the initiation stage.
  5. In figure 1 presented flow of the participants not very clear: there is some additional dropouts of participants, but still 16 participants left (6 and 7 steps).
  6. Why added some text with some explanations to Material & Methods part (line 162-168).
  7. It is not clear how was calculated or taken MVPA time (MVPA* (minutes per day).
  8. In Table 2 presented measures, but it is not clear which data was taken here (1st testing and the last). I would suggest including in the table descriptive data.
  9. To short in my opinion refence list for such an article.
  10. Additionally, discussion part could be improved as well.

Before publication, in my opinion, article must be improved.

Author Response

Manuscript ID: sustainability-1214606

Title: Initial Positive Indications with Wearable Fitness Technology Followed by Relapse: What’s going on?

We would like to take this opportunity to thank reviewers for valuable and relevant comments that have increased the overall quality of our revised manuscript. Below you will find our suggestion for feedback on comments made. We hope that our answers satisfy your critical comments and that you consider accepting our revised manuscript.

Sincerely,

the authors

Reviewer’s comments

Authors’ Responses

The title of the article is not correctly formulated.

Thank you for your comment, would you please guide us as to how to correctly formulate the title? Thank you

Some terms usage could be corrected, e.g., “physical capacity” etc.

Line 25. Replaced “capacity” with “health”

I missed some anthropometric data in this research (e.g., body weight, height, BMI etc.) it would be more informative and additionally for trying to find some links to the motivation. It would be possible to add this information, because in apple watches this information necessary at the initiation stage.

Line 106-108: We agree that some anthropometric data can enrich the description of the sample group and  have included the initial BI of all subjects “16 participants (male = 5; female = 11) with a mean age of 51 years (SD = 8.4) and body mass index (BMI) of 28.23 ( SD = 3.79); men 45 years (SD = 10.4) and BMI of 31.02  (DS = 2.54)  and; women 53.7 (SD = 6.1) and BMI of 27.00 (SD = 3.67)”

After some deliberation we chose not to perform further analyses of this data as this would increase the number of variables analyzed in a small sample size increases the risk for statistical error and because the aim of the study was to motivate participants to increase the PA behavior and not weight reduction.

In figure 1 presented flow of the participants not very clear: there is some additional dropouts of participants, but still 16 participants left (6 and 7 steps).

Thank you for highlighting this. The original investigation design included physical tests in which 2 participants were unable to participate. However, this is irrelevant for the current study and we have amended figure 1.

Why added some text with some explanations to Material & Methods part (line 162-168).

Line 162-168: This was a formatting error and the text has been deleted

It is not clear how was calculated or taken MVPA time (MVPA* (minutes per day).

Line 161: MVPA is calculated through an inbuilt algorithm in the apple watch, we downloaded this data from the Health app and have corrected this in line 147 by deleting “exercise time” and replaced it with “MVPA time”.

In Table 2 presented measures, but it is not clear which data was taken here (1st testing and the last). I would suggest including in the table descriptive data.

We have included Table 4 Descriptive Statistics for Motivation Measures. We believe that this table in combination with Table 3 clearly shows which data is used in the repeated measure ANOVA.

To short in my opinion refence list for such an article.

We have added the following references:

5. Tester, J., & Baker, R.. Making the playfields even: Evaluating the impact of an environmental intervention on park use and physical activity. Preventive Medicine. 2009 48 (4), 316-320.

6. Kelly, B, & Fry, J. Camden outdoor gyms evaluation: Phase 1 [document on the internet] London:London Borough of Camden 2010;[cited 2019 Jan 21], Available from: https://opendata.camden.gov.uk/Leisure/Pro-Active-Camden-Outdoor-Gym Evaluation-2011/qzhz-a3y6

7. Barton J, Pretty J. What is the best dose of nature and green exercise for improving mental health? A multi-study analysis Environ Sci Technol. 2010, 44:3947–55. doi:10.1021/es903183r

10. Bice, M.R., Ball, J.W., & McClaran, S. Technology and physical activity motivation. J. Sport Exerc. Psychol. 2016, 14 (4), 295-304.

11. Bol, N., Helberger, N., & Weert, J. C. M. Differences in mobile health app use: A source of new digital inequalities? J. Soc. Inf. Disp. 2018,, 34(3), 183–193. doi:10.1080/01972243.2018.1438550

 28. Cranney, L., Phongsavan, P., Kariuki, M., Stride, V., Scott, A., Hua, M., & Bauman, A . Impact of an outdoor gym on park users’ physical activity: A natural experiment. Health Place. 2016, 37, 26–34. doi:10.1016/j.healthplace.2015.11.002

Additionally, discussion part could be improved as well.

Line 264-279: we have improved the discussion by including a general view of the topic

Reviewer 3 Report

An interesting article on the use of digital technology to promote physical activity. I have some comments and suggestions. 

The authors in the introduction or the discussion should talk about the issue of digital inequalities. From a public health perspective, I don't know if these technologies are accessible to everyone and what the population impact is. 

Specific comments: 
Line 137: There is one period too many before the term Procedure. 
Lines 185 and 188: In tables 2 and 3, would it be possible to add the n= (number of participants in the analyses). 
Line 194: There is one space too many in the sentence. 
Line 253: there is one period too many at the end of the sentence. 
For the discussion section, would it be possible to add a few sentences about the strengths of the study. 

Translated with www.DeepL.com/Translator (free version)

Author Response

Manuscript ID: sustainability-1214606

Title: Initial Positive Indications with Wearable Fitness Technology Followed by Relapse: What’s going on?

We would like to take this opportunity to thank reviewers for valuable and relevant comments that have increased the overall quality of our revised manuscript. Below you will find our suggestion for feedback on comments made. We hope that our answers satisfy your critical comments and that you consider accepting our revised manuscript.

Sincerely,

the authors

Reviewer

Reviewer’s comments

Authors’ Responses

The authors in the introduction or the discussion should talk about the issue of digital inequalities. From a public health perspective, I don't know if these technologies are accessible to everyone and what the population impact is.

Line 48-53: now reads “Activity devices are considered financially economical for research because of the reasonable cost but may not be equally accessible to the general population such as those with a lower social economic background [11]. One solution to cater for this inequality is the activity trackers could be prescribed by a doctor as an economically viable option and successful component in healthcare interventions.”

Line 137: There is one period too many before the term Procedure. 

Line 137: Deleted the extra period

Lines 185 and 188: In tables 2 and 3, would it be possible to add the n= (number of participants in the analyses). 

Line 195 & 197: We have included “number of participants = 16”

Line 194: There is one space too many in the sentence. 

Line 194: Deleted the extra space

Line 253: there is one period too many at the end of the sentence. 

Line 253: Deleted the extra period.

For the discussion section, would it be possible to add a few sentences about the strengths of the study. 

Line 264-279 we have improved the discussion by including a general view of the topic and a few sentences on the strengths of the current study.